# LEARNING TO COMPLETE CODE WITH SKETCHES

**Daya Guo**
School of Computer Science and Engineering
Sun Yat-sen University, China
guody5@mail2.sysu.edu.cn

**Alexey Svyatkovskiy**
Microsoft
Redmond, WA, USA
alsvyatk@microsoft.com

**Jian Yin**
School of Computer Science and Engineering
Sun Yat-sen University, China
issjyin@mail.sysu.edu.cn

**Nan Duan**
Microsoft Research
Beijing, China
nanduan@microsoft.com

**Marc Brockschmidt, Miltiadis Allamanis**
Microsoft Research
Cambridge, UK
{mabrocks,miallama}@microsoft.com

## ABSTRACT

Code completion is usually cast as a language modelling problem, *i.e.*, continuing an input in a left-to-right fashion. However, in practice, some parts of the completion (*e.g.*, string literals) may be very hard to predict, whereas subsequent parts directly follow from the context. To handle this, we instead consider the scenario of generating code completions with "holes" inserted in places where a model is uncertain. We develop GRAMMFORMER, a Transformer-based model that guides code generation by the programming language grammar, and compare it to a variety of more standard sequence models.

We train the models on code completion for C# and Python given partial code context. To evaluate models, we consider both ROUGE as well as a new metric REGEXACC that measures success of generating completions matching long outputs with as few holes as possible. In our experiments, GRAMMFORMER generates 10-50% more accurate completions compared to traditional generative models and 37-50% longer sketches compared to sketch-generating baselines trained with similar techniques.

## 1 INTRODUCTION

Recent high-capacity language models (LM) have shown that machine learning models are able to generate coherent, realistic text, but it is often hard to guide them towards a specific goal, especially when describing the intent is complex or more costly than manually generating the target output.

One such scenario are LMs of source code (LMC). Since Hindle et al. (2012) increasingly sophisticated LMCs have been built, including transformer-based ones, such as those of Svyatkovskiy et al. (2020); Feng et al. (2020); Chen et al. (2021) and various similar unpublished models such as TabNine and SourceAI. These models generate full sequences of code tokens left-to-right with any prefix acting as the (partial) user intent. While LMs generate realistic-looking outputs, they are known to occasionally "hallucinate" (Puduppully et al., 2019; Malmi et al., 2019; Maynez et al., 2020; Liu et al., 2021), *i.e.* generate plausible but incorrect content. This is particularly problematic in generating source code, where small mistakes can lead to erroneous code that is very hard to debug or introduces vulnerabilities (Pearce et al., 2021).

In this work, we investigate models that can decline to make predictions in places where there is high uncertainty (*e.g.*, where the user should choose a name), but continue generating around these "holes". For example, in Fig. 1(left) a developer has typed some code and is about to type the next line. A likely completion is to consume more command line arguments, but their name is unclear

**Code Context**:

```
1 import sys
2 target = sys.argv[1]
3 I
```

**Ground-Truth**:
```
ID = sys.argv[2]
```

**Suggested Code Completions**:

| | |
|---|---|
| $L \to R$ | `target = target.replace("\\", "/")` |
| $L \to R + \varnothing$ | `target =` |
| $L \to R + \blacksquare$ | `print(target)` |
| Copilot | *(No suggestion)* |
| GRAMMFORMER | `■ = sys.argv[2]` |

Figure 1: A sample snippet (left; abbreviated from Fig. 12 in Appx. A). A developer has just typed the code and their cursor (in blue) is at line 3. Code completions provided by a number of models are shown on the right, where $L \to R$ is a standard LMC and GRAMMFORMER is our new model.

from the context. A traditional generative model (*e.g.* Fig. 1; top right) may choose to provide a completion that exists in the training data, but is not clearly called for here. On the other hand, a model able to explicitly mark where it is uncertain (Fig. 1; bottom right) makes it clear to a user where further input is required.

However, creating such models is not trivial. A simple first attempt may be to use a standard LMC, but output a "hole token" ■ whenever the model is uncertain about the next output token. However, continuing after the "■" then becomes infeasible, as the LMC was not trained on such data. Hence, a suitable training dataset and objective need to devised. As no large datasets with holes exist, we instead choose to use a reinforcement learning approach in which our reward function encourages the model to make "long" predictions with as few "■" tokens as possible, but to avoid making incorrect predictions. We found that standard left-to-right sequence models perform poorly on this task. Hence, we developed GRAMMFORMER, a model that construct suggestions by generating a (partial) syntax tree, but which has the option of leaving non-terminals in its output.

**Contributions** (1) We present GRAMMFORMER, a transformer-based model that generates code based on the programming language grammar and can predict hole tokens rather than output it is uncertain about. (2) We develop REGEXACC, a metric that evaluates the quality of predictions with holes. (3) We evaluate GRAMMFORMER on Python and C# code and show that GRAMMFORMER makes longer and more precise statement-level sketch completions compared to baselines.

## 2 METHOD

Our aim is to predict code completions as *sketches*, a mix of actual tokens and "holes" ■, which are meant to signify that the model is unable to make a useful prediction within the given context and further user input is required. Formally, we consider models that take a context sequence **x** of tokens as input and have to produce an output sequence **y**; intuitively, **x** is what the user typed so far, and **y** is the suggestion presented to the user. In our setting, **y** is a *sketch*, a mix of tokens from the programming language and the special token ■ signifying a "hole" that could be filled by an arbitrary sequence of tokens. For example, `t = foo(■)` is a sketch corresponding to assigning the return value of function `foo` to variable `t`, but leaves the arguments of the function call undetermined.

**Metric** A good sketch is one that (a) can be completed into the correct output and (b) is as precise as possible. To measure how successful a method is in doing so, we define a new metric REGEXACC. For (a), we use `toRegex(ŷ)` to turn a predicted code sketch $\hat{y}$ into a regular expression by replacing all holes with the wildcard matching any non-empty sequence (".+" in Perl Compatible Regular Expression syntax). If the regex matches the ground truth, `matches(·,·)` returns a score of 1 otherwise it returns 0. To implement (b), we scale this result by the proportion of terminal tokens predicted, by defining `nTokens(ŷ)` as the function that returns the number of non-hole symbols in $\hat{y}$. More formally, assume an output sketch $\hat{y}$ and a ground-truth sequence $y^*$, where $y^*$ does *not* contain any ■ tokens. REGEXACC is then defined as

$$\text{REGEXACC}(\hat{y}, y^*) \triangleq \texttt{matches}(\texttt{toRegex}(\hat{y}), y^*) \cdot \frac{\texttt{nTokens}(\hat{y})}{\texttt{nTokens}(y^*)}.$$

Beyond REGEXACC, we also consider ROUGE (Lin, 2004), since a sketch can be thought as a form of a "summary" of the target text. For this, we use a helper function ERASEHOLES($\hat{y}$) that simply

| | | | | | | | | | | |
|---|---|---|---|---|---|---|---|---|---|---|
| $\mathbf{x}^{(0)}$: | r = | ⟨Expr⟩ | | | | | | | | $i^{(0)} = 3$ |
| $\mathbf{x}^{(1)}$: | r = | ⟨Expr⟩ | * | ⟨ParenthesizedExpr⟩ | | | | | | $i^{(1)} = 5$ |
| $\mathbf{x}^{(2)}$: | r = | ⟨Expr⟩ | * | ( | ⟨Expr⟩ | | | ) | | $i^{(2)} = 6$ |
| $\mathbf{x}^{(3)}$: | r = | ⟨Expr⟩ | * | ( | ⟨Expr⟩ | – | ⟨Expr⟩ | ) | | $i^{(3)} = 8$ |
| $\mathbf{x}^{(4)}$: | r = | ⟨Expr⟩ | * | ( | ⟨Expr⟩ | – | ⟨Identifier⟩ | ( ⟨ArgList⟩ ) | ) | $i^{(4)} = 8$ |
| $\mathbf{x}^{(5)}$: | r = | ⟨Expr⟩ | * | ( | ⟨Expr⟩ | – | foo | ( ⟨ArgList⟩ ) | ) | $i^{(5)} = 10$ |
| $\mathbf{x}^{(6)}$: | r = | ⟨Expr⟩ | * | ( | ⟨Expr⟩ | – | foo | ( ⟨Identifer⟩ ) | ) | $i^{(6)} = 10$ |
| $\mathbf{x}^{(7)}$: | r = | ⟨Expr⟩ | * | ( | ⟨Expr⟩ | – | foo | ( args ) | ) | $i^{(7)} = 6$ |
| $\mathbf{x}^{(8)}$: | r = | ⟨Identifier⟩ | * | ( | ⟨Expr⟩ | – | foo | ( args ) | ) | $i^{(8)} = 6$ |
| $\mathbf{x}^{(9)}$: | r = | x | * | ( | ⟨Expr⟩ | – | foo | ( args ) | ) | $i^{(9)} = \oslash$ |

Figure 2: Progress of grammar-based code generation of the sketch `r = x * (■-foo(args))` by GRAMMFORMER. Each line represents consecutive $\mathbf{x}^{(t)}$ in Alg. 1. Terminal tokens are shown in `monospace blue` font. The underlined non-terminal at position $i^{(t)}$ is selected by $P_s$ and its expansion is generated by $P_e$, *i.e.* the output underneath the selected (underlined) non-terminal. Fig. 5 and Fig. 6 in Appx. A show real example generation sequences from our datasets.

drops all ■ tokens, and then consider $\text{ROUGE}_{\text{F1}}(\text{ERASEHOLES}(\hat{\mathbf{y}}), \mathbf{y}^*)$. ROUGE is more lenient to errors than REGEXACC and gives partial credit to non-matching but plausible sketches.

## 2.1 LINEAR CODE SKETCH GENERATION

First, we consider the idea of generating code sketches using a standard generative model for language. To this end, we simply extend the vocabulary with the special "■" token. An obvious problem is that while we have plenty of training data for a standard generative model, we do not have training data for outputs $\mathbf{y}$ that contain the ■ token. Consequently, we cannot train the model in a fully supervised fashion, and instead turn to reinforcement learning. Concretely, we devise a reward function $r(\cdot)$ that averages REGEXACC and ROUGE, *i.e.* for a predicted output sketch $\hat{\mathbf{y}}$ and a ground truth output (without ■ tokens) $\mathbf{y}^*$, we define

$$r(\hat{\mathbf{y}}, \mathbf{y}^*) = \frac{1}{2}\left(\text{REGEXACC}(\hat{\mathbf{y}}, \mathbf{y}^*) + \text{ROUGE}_{\text{F1}}(\text{ERASEHOLES}(\hat{\mathbf{y}}, \mathbf{y}^*))\right). \tag{1}$$

Using the combination of ROUGE (which does not consider holes) and REGEXACC is crucial here, as ROUGE is much "smoother" compared to REGEXACC, which is 0 for all but very few predictions, allowing us to measure partial improvement. We use our reward function from Eq. 1 to evaluate the quality of the output of the full model and compute a loss. Inspired by Paulus et al. (2017) we use self-critical policy gradient training (Rennie et al., 2017) and for a prediction $\hat{\mathbf{y}}$ we minimise

$$\mathcal{L}(\mathbf{x}, \mathbf{y}^*) = \left(r(\hat{\mathbf{y}}, \mathbf{y}^*) - \tilde{r}(\mathbf{x})\right) \cdot \mathcal{L}_{\text{gen}}(\mathbf{x}, \hat{\mathbf{y}}) \tag{2}$$

Here, $\tilde{r}(\mathbf{x})$ is the reward achieved by the prediction from the snapshots of the model that achieved the best score so far and $\mathcal{L}_{\text{gen}}$ is the loss of the generative model. Intuitively, this objective rewards models that improve upon the previous best policy with respect to $r$.

To model this in practice, we use a standard encoder/decoder Transformer model Vaswani et al. (2017); Radford et al. (2019), "translating" the context $\mathbf{x}$ into the output $\mathbf{y}$ using separate encoder and decoder models. We additionally also consider the language modelling case, *i.e.*, a model that conditioned on $\mathbf{x}$ predicts token $\mathbf{y}_0$, conditioned on $\mathbf{x}, \mathbf{y}_0$ predicts token $\mathbf{y}_1$, *etc.*.

**Pretraining** In practice, we found that directly training a sequence model to maximise Eq. 1, is very slow and does *not* converge to a useful model. Instead, we heuristically generate a dataset suitable for supervised pretraining. We replace random AST non-terminals of the target output by ■ and generate target sequences. These contain terminals and zero or more ■. We then pretrain the model on this dataset to convergence, and then fine-tune it using the reward of Eq. 1.

## 2.2 GRAMMAR-BASED CODE SKETCH GENERATION

In experiments, we found the simple extended sequence model from above to not perform well, in particular, ■ tokens would not replace semantically meaningful subsequences (*e.g.* "`szconv.■`)"

---

**Algorithm 1** GRAMMFORMER generative process, given an input sequence $\mathbf{x}^{(0)}$.

**for** $t = 0, 1, 2, ...$ **do**

    $i^{(t)} \sim P_s\left(i \mid \mathbf{x}^{(t)}, N(\mathbf{x}^{(t)})\right)$          ▷ sample non-terminal position from $N(\mathbf{x}^{(t)})$ to expand

    **if** $i^{(t)} = \oslash$ **then**          ▷ if $\mathbf{x}^{(t)}$ does *not* contain non-terminals or none was selected by $P_s$

        **break**          ▷ stop generation

    $\mathbf{u}^{(t)}_{\oplus i^{(t)}} \sim P_e\left(\mathbf{u} \mid \mathbf{x}^{(t)}, i^{(t)}\right)$          ▷ sample expansion of non-terminal at position $i^{(t)}$

    $\mathbf{x}^{(t+1)} \leftarrow \mathbf{x}^{(t)}_{<i^{(t)}} :: \mathbf{u}^{(t)}_{\oplus i^{(t)}} :: \mathbf{x}^{(t)}_{>i^{(t)}}$     ▷ create $\mathbf{x}^{(t+1)}$ by replacing non-terminal at $i^{(t)}$ by $\mathbf{u}^{(t)}_{\oplus i^{(t)}}$

**return** NONTERMINALSTOHOLES$(\mathbf{x}^{(t)})$ ▷ convert remaining non-terminals to holes and return

---

does not contain a left parenthesis and requires the user to fill it in.). To resolve this, we developed GRAMMFORMER, a *grammar-guided* model. It generates code by following the structure of the context-free grammar (CFG) defining the programming language syntax, iteratively expanding non-terminal symbols. Crucially, it can choose to *not* expand some non-terminal symbols, which can then be presented as ■ to users. In traditional grammar-based generation of text (Cohen et al., 2012) or code (Maddison & Tarlow, 2014; Yin & Neubig, 2017; Allamanis & Sutton, 2014; Bielik et al., 2016), the CFG is followed by sequentially expanding the left-most, bottom-most non-terminal symbol, using one of the production rules of the grammar. GRAMMFORMER changes this and instead selects the non-terminal symbol to expand, if any. An example generation is shown in Fig. 2.

**Probabilistic Model** A CFG is defined as a tuple $(\Sigma, \mathcal{N}, S, R)$ where $\Sigma$ is a set of terminal symbols, $\mathcal{N}$ is a set of non-terminal symbols, $S \in \mathcal{N}$ is the root symbol and $R$ is a set of production rules. We denote non-terminals as ⟨NonTerminalName⟩. GRAMMFORMER can be viewed as a sequence-to-sequence model transforming $\mathbf{x} = \mathbf{x}_0, \mathbf{x}_1, ..., \mathbf{x}_n$ into a new sequence in which one non-terminal symbol $x_i$ has been replaced by a new sequence of new symbols, according to a production rule of the grammar. Examples of such sequences and rewrites are shown in Fig. 2.

GRAMMFORMER does this rewriting in two steps. First, a *non-terminal selector model* $P_s$ selects a non-terminal in $\mathbf{x}$ to expand and then the *non-terminal expansion model* $P_e$ determines how to expand it. To define $P_s$, let $N(\mathbf{x}) = \{i \mid \mathbf{x}_i \in \mathcal{N}\} \cup \{\oslash\}$ denote the set of non-terminal positions in $\mathbf{x}$ and a special "stop expansion" $\oslash$ symbol. Conditioned on $\mathbf{x}$, $P_s$ produces a probability distribution over $N(\mathbf{x})$. In turn, $P_e$ is conditioned on $\mathbf{x}$ and a position $i \in N(\mathbf{x})$ and models a probability distribution over expansion sequences $\mathbf{u} \in (\Sigma \cup \mathcal{N})^*$. Note that factorising GRAMMFORMER into two models $P_s$ and $P_e$ is an important modelling decision: how to best expand a non-terminal is entirely separated from predicting whether a hole should be introduced. These two concepts are intermixed in standard (sequence) decoders. In practice, we define both models using neural architectures with partially shared parameters, as discussed below.

Alg. 1 shows a high-level description of GRAMMFORMER, in which $P_s$ and $P_e$ are used repeatedly to select and expand non-terminals (not necessarily the left-most one), until none are left or the $P_s$ indicates that expansion should stop. Here, NONTERMINALSTOHOLES$(\cdot)$ replaces all remaining non-terminal symbols with a hole ■. Note that GRAMMFORMER is *not* context-free, taking into account the whole input sequence when expanding a non-terminal. Second, in contrast to many grammar-based methods (Yin & Neubig, 2017; Bielik et al., 2016), any non-terminal can be expanded at each step. Finally, $P_e$ is not directly constrained to follow the production rule set $R$, but can generate any sequence. In practice, it learns to follow to the rules of $R$ from the data, but this flexibility is important for handling string literals and argument tuples of variable length.

**Neural Model** To implement $P_s$ and $P_e$, we use a shared encoder module that computes a representation of the input sequence $\mathbf{x} = \mathbf{x}_0, \ldots, \mathbf{x}_n$ as vectors $\mathbf{e}_0, \ldots, \mathbf{e}_n$, $\mathbf{e}_i \in \mathbb{R}^D$, where $D$ is a hyperparameter. Our encoder module is a Transformer (Vaswani et al., 2017), given the impressive results of transformer-based models in NLP and code (Feng et al., 2020). Other architectures (RNNs, 1D-CNNs, Transformer variants) would be suitable, but we leave their study for future work.

$P_s$ is implemented similar to a pointer network on top of this encoder module, i.e.

$$P_s(i \mid \mathbf{x}) = \underset{i \in N(\mathbf{x})}{\text{softmax}}\left(f(\mathbf{e}_i)\right),$$

where $f$ is a learnable feed-forward neural network. For our purposes, we define $\mathbf{e}_{\oslash}$ as the representation of the special start symbol `[CLS]` used in our Transformer encoder.

The expansion model $P_e$ follows a standard autoregressive decoder formulation, *i.e.*

$$P_e(\mathbf{u} \mid \mathbf{x}, i) = \prod_{j=1}^{m} P_{dec}(\mathbf{u}_j \mid \mathbf{e}_0, \dots, \mathbf{e}_n, i, \mathbf{u}_{<j}).$$

We implement $P_{dec}$ as a (causal) relational Transformer decoder, similar to Wang et al. (2019). Relational transformers augment the attention mechanism by incorporating predefined relationships among elements; attention scores are then biased by learnable weights for each relation. In GRAMM-FORMER, we only use a single relation, connecting each token to the expanded non-terminal token $\mathbf{x}_i$, to help the model focus on the token it needs to generate an expansion for.

**Objective**   Due to the lack of supervised data, we employ reinforcement learning to train GRAMM-FORMER. We use our reward function from Eq. 1 to evaluate the quality of the output of the full model. We use self-critical policy gradient training as in Eq. 2 and minimise

$$\mathcal{L}(\mathbf{x}, \mathbf{y}^*) = \left(r(\hat{\mathbf{y}}, \mathbf{y}^*) - \tilde{r}(\mathbf{x})\right) \cdot \sum_{t=0}^{T} \left(-\log P_s\left(i^{(t)} \mid \mathbf{x}^{(t)}\right) - \mathbb{I}\left(i^{(t)} \neq \oslash\right) \cdot \log P_e\left((\mathbf{u}_{\oslash i^{(t)}}^{(t)})^* \mid \mathbf{x}^{(t)}, i^{(t)}\right)\right).$$

(3)

Here, $\tilde{r}(\mathbf{x})$ is the reward achieved by the snapshots of $P_s$ and $P_e$ that achieved the best score so far. The rest of the objective follows the iterations of the loop in Alg. 1, where $t$ is the iteration index, $\hat{\mathbf{y}}$ is the predicted sketch, $\mathbf{y}^*$ is the ground-truth sequence of terminals, and $\mathbb{I}(\cdot)$ is the indicator function.

**Pretraining**   As in the sequence model, directly training with the RL objective Eq. 3 is computationally intensive due to the sampling requirement. We again use a pretraining strategy. First, we train $P_e$ to expand every non-terminal, independently of the expansion order learned by $P_s$. To do this, we use the input training examples and follow Alg. 1, but instead of sampling from $P_s(\cdot)$, we sample $i^{(t)}$ from a uniform distribution over the non-terminals in $\mathbf{x}^{(t)}$, $\widetilde{N}(\mathbf{x}^{(t)}) = \{i \mid \mathbf{x}_i \in \mathcal{N}\}$. This yields sequences of intermediate sketches $\mathbf{x}^{(t)}$ for each example. Furthermore, for each $\mathbf{x}^{(t)}$, we compute the ground-truth expansion $(\mathbf{u}_{\oslash i}^{(t)})^*$ for all non-terminals $i \in \widetilde{N}(\mathbf{x}^{(t)})$. We can then pretrain $P_e$ using the supervised objective

$$\mathcal{L}_{\text{pre, e}}\left(\mathbf{x}^{(t)}, (\mathbf{u}_{\oslash i}^{(t)})^*_{i \in \widetilde{N}(\mathbf{x}^{(t)})}\right) = \frac{1}{|\widetilde{N}(\mathbf{x}^{(t)})|} \cdot \sum_{i \in \widetilde{N}(\mathbf{x}^{(t)})} -\log P_e\left((\mathbf{u}_{\oslash i^{(t)}}^{(t)})^* \mid \mathbf{x}^{(t)}, i\right),$$

*i.e.* the negative log-likelihood of the correct expansion for *all* non-terminals in $\mathbf{x}^{(t)}$. This computation is more computationally efficient compared to the one in Eq. 3 since the cost of encoding $\mathbf{x}^{(t)}$ is amortised across all potential expansions and no sampling is required. Once $P_e$ is pretrained, we pretrain $P_s$. For this, we fix the weights of the shared encoder module, and optimise only the remaining parameters of $P_s$ through Eq. 3. Once we have a pretrained both models, we then fine-tune all model weights end-to-end, using Eq. 3.

**Optimisation: Grammar Flattening**   Following the formal grammar of a programming language commonly introduces tedious expansions. For example, the Python non-terminal $\langle\mathsf{Call}\rangle$ is always expanded to $\langle\mathsf{Expr}\rangle(\langle\mathsf{ArgumentList}\rangle)$, and the C# non-terminal $\langle\mathsf{NotEqualOp}\rangle$ is always expanded to the terminal `!=`. We "flatten" the grammar by replacing non-terminals such as $\langle\mathsf{Call}\rangle$ and $\langle\mathsf{NotEqualOp}\rangle$ with all their possible expansions. In Appx. C we provide the list of the flattened non-terminals. Note that if we repeated this process for all non-terminals except from the starting symbol $S$, GRAMMFORMER would degenerate into a standard encoder-decoder model.

**Beam Search**   At test time, we employ a two-step beam search, and replace sampling from $P_s$ and $P_e$ with their top-$\nu$ outputs, keeping a beam of size $k$. First, for each $\mathbf{x}^{(t)}$ in the beam, we compute $P_s$ and select the top-$m$ non-terminal positions to expand. For each of those $m$ positions, we sample the top-$n$ expansions from $P_e$ using a standard beam search. We compute the likelihood of all $k \cdot n \cdot m$ results, and then keep only the top-$k$. This process (detailed in Appx. E) is similar to a standard beam search but takes into account that two submodels are used.

**Computational Cost** GRAMMFORMER's ability to predict sketches comes with additional computational cost compared to standard transformer encoder-decoders: at each iteration of the loop in Alg. 1 $\mathbf{x}^{(t)}$ changes, $P_s$ and $P_e$ must be recomputed. This means that the encoder-decoder runs once on each partial sequence, in contrast to left-to-right causal generation, in which intermediate results can be re-used. Future work may consider selecting more than one element to expand from $N(\mathbf{x}^{(t)})$ at each step, reducing the expansion steps, similar to Welleck et al. (2019); Stern et al. (2019).

## 3 EVALUATION

To empirically evaluate our model's ability to predict useful completions, we use REGEXACC and ROUGE. Note that we measure these on the generated sequence, *i.e.* we ignore the context tokens.

**Datasets** To collect a dataset, we clone all non-fork repositories with more than 20 stars on GitHub that have C# or Python as their top language. Then, we deduplicate the corpus using the method of Allamanis (2019); Lopes et al. (2017). Finally, we parse all files into a syntax tree using Tree-sitter, ignoring any files that cannot be parsed using the v0.19.0 grammar definitions. Finally, we split the files into 70-10-20 train-validation-test. To create (pre-)training examples, *i.e.* inputs to Alg. 1, we search the syntax tree of each file and for each ⟨SimpleStatement⟩ non-terminal create an example. The syntax tree rooted at the ⟨SimpleStatement⟩ non-terminal is then used to get the ground-truth expansions during pre-training and the ground-truth expansion $\mathbf{y}^*$. For our test set, we randomly sample a ⟨SimpleStatement⟩ non-terminal for each file to evaluate and obtain 318K (resp. 362K) examples for C# (resp. Python). For each example, $\mathbf{x}$ is the 200 terminal tokens *before* the ⟨SimpleStatement⟩ non-terminal. More details about the dataset can be found in Appx. B.

**Baselines** Since we are not aware of any prior model that targets code completion with sketches, we consider two Transformer-based baselines. We consider both the sequence-to-sequence setting using separate encoder and decoder models (Vaswani et al., 2017) as well as the language modelling setting (where there is no distinction between encoder and decoder). We refer to these as "$L \rightarrow R$" and "$LM$". We use "$L \rightarrow R$" to denote a standard Transformer encoder-decoder model (Vaswani et al., 2017) used in sequence-to-sequence tasks. Additionally, we consider "$L \rightarrow R + \oslash$" and "$LM + \oslash$", which are trained to stop generation by inserting a final ■ token that captures any suffix. Note that this models can only generate sketches that are prefixes of the target completion, *i.e.* it corresponds to a standard token-level generative model with a learnable stopping ability. To train this model, we use self-critical policy gradient training (as in Eq. 2).

**Model Training** We provide the training details for all experiments. Most of our models use a 6-layer Transformer as encoder and 6-layer Transformer as decoder, each with a hidden dimension of 768 and 12 attention heads, with the exception of the $LM$ model (and its variations), which uses a single 12-layer Transformer, to match the number of parameters of the other models. We set the intermediate dimension of each Transformer layer as 3072 and use 3 fully-connected layers with 3072, 768 and 1 hidden sizes as the feed-forward neural network $f$ in the selector model $P_s$. The vocabulary is constructed using byte-pair encoding (Sennrich et al., 2015) and the vocabulary size is 25 000. We set max length of input and output sequences as 512 and 64, respectively. We train the model with Adam optimiser using a learning rate of 2e-5 and batch size 4 096. We used automatic mix precision. Training used 64 NVIDIA Tesla P100 with 16GB memory for 10 days. For beam search we use $k = 5, n = 1$ and $m = \infty$, *i.e.* we consider all non-terminals in each $\mathbf{x}^{(t)}$. We selected these during early experiments as a reasonable trade-off between speed and predictive performance.

**Results** Tbl. 1 shows the results for all considered models. For both Python and C#, GRAMMFORMER outperforms the baseline methods in terms of REGEXACC, showing that the grammar-based generation can create better sketches compared to simpler methods. Note that although $L \rightarrow R$ has a comparable or better ROUGE score, it does substantially worse than GRAMMFORMER with respect to REGEXACC, meaning that the predictions are "similar" but the sketches contain errors (*i.e.* do not match the ground-truth). This means that if a code completion system suggested the full output of $L \rightarrow R$, the user would have to pause and correct the suggestion more frequently. On the other hand, $L \rightarrow R + \oslash$ improves over $L \rightarrow R$ in terms of REGEXACC but has a worse ROUGE and generates significantly shorter suggestions (5.3 *vs.* 7.5 tokens-long for C#). This is expected since

Table 1: Performance of GRAMMFORMER compared to baselines for Python and C#.

| | C# | | | | Python | | | |
|---|---|---|---|---|---|---|---|---|
| | REGEXACC | | ROUGE | Avg | REGEXACC | | ROUGE | Avg |
| | Top 1 | Top 5 | | Len | Top 1 | Top 5 | | Len |
| $LM$ | 0.42 | 0.52 | 75.7 | **8.0** | 0.18 | 0.24 | 51.0 | **8.6** |
| $L \rightarrow R$ | 0.42 | 0.47 | 77.0 | 7.1 | 0.17 | 0.20 | **53.2** | 5.8 |
| $LM + \oslash$ | 0.42 | 0.49 | 70.9 | 6.8 | 0.19 | 0.25 | 49.5 | 7.3 |
| $L \rightarrow R + \oslash$ | 0.45 | 0.54 | 69.1 | 5.3 | 0.20 | 0.29 | 39.3 | 3.0 |
| $LM + \blacksquare$ | 0.44 | 0.54 | 73.3 | 6.3 | 0.20 | 0.27 | 53.9 | 6.6 |
| $L \rightarrow R + \blacksquare$ | 0.45 | 0.55 | 73.5 | 5.8 | 0.18 | 0.22 | 48.9 | 4.7 |
| GRAMMFORMER (pre-trained only) | 0.45 | 0.57 | 77.0 | 7.2 | 0.20 | 0.29 | 50.2 | 5.7 |
| GRAMMFORMER | **0.47** | **0.59** | **77.4** | 7.5 | **0.21** | **0.30** | 51.6 | 6.1 |

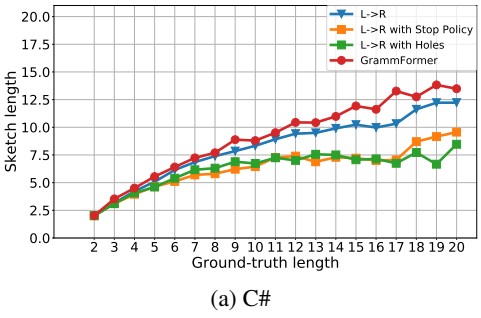
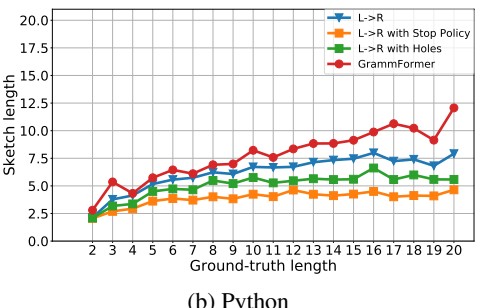

(a) C#        (b) Python

Figure 3: Sketch length *vs.* ground-truth length

$L \rightarrow R + \oslash$ is trained to be more "conservative" (*i.e.* avoid incorrect suggestions) but is also unable to introduce holes beyond the last generated token. Finally, we can see that while GRAMMFORMER already performs well after our pre-training procedure, we can further improve its performance with our fine-tuning technique. We believe that this is because when $P_s$ and $P_e$ are trained jointly, they co-adapt: some of the capacity of the shared encoder module that is used to make predictions for hard-to-expand non-terminals is "freed" since $P_s$ learns to not expand them.

Fig. 3 shows how the length of the generated code sketch relates to the length of the ground truth expression. While the differences between models are small for short target sequences, GRAMM-FORMER generates substantially longer suggestions than other models when more complex suggestions are required. In particular, the $L \rightarrow R + \oslash$ model generates very short suggestions, as it is trained to stop generation whenever it reaches a point at which it is uncertain about the next token.

Fig. 4 in turn shows how often the suggested sketch was correct dependent on the length of the ground truth token sequence. Here, $L \rightarrow R + \oslash$ does best because it generates the shortest (*i.e.*, least determined) predictions, which is exactly the trade-off captured by our REGEXACC metric. Of the models that generate longer suggestions, GRAMMFORMER clearly does best, with the improvement becoming more pronounced with the length of the target sequence. Note that the performance of the models on C# is generally better compared to the performance in Python. We believe that this has to do with the grammar of each language and the patterns it induces within the developer's code. Casalnuovo et al. (2019); Karampatsis et al. (2020) have observed a similar phenomenon on the perplexity across (standard left-to-right) language models for different programming languages.

**Ablations** Next, we look into ablations of GRAMMFORMER and reason about how its components perform. To this end, Tbl. 2 shows the performance of different model variants on the C# dataset. First, we analyse the effect of the selector model $P_s$. To this end, we consider two ablations. The first is the "random expansion" model, in which the non-terminal token to expand is sampled uniformly at random from the full set of non-terminal symbols, and which hence does not stop expansion as long as any holes are remaining. This is effectively GRAMMFORMER after our pre-training procedure for $P_e$. This model achieves the best ROUGE score, but a relatively bad REGEXACC, as it is forced to generate a prediction even when it is very uncertain. The second ablation, a "fixed threshold"

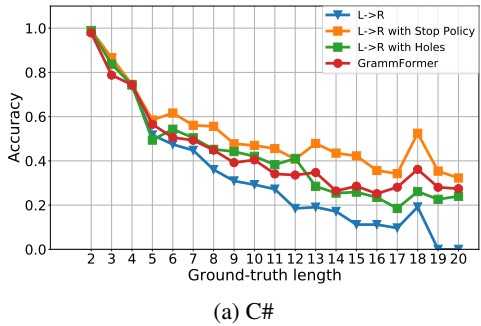

(a) C#

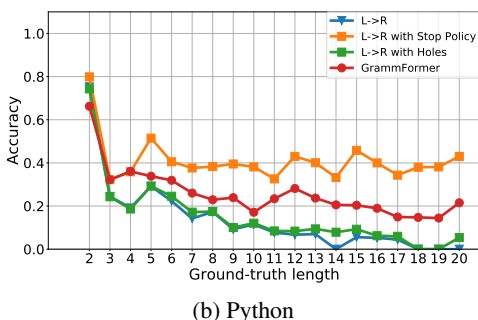

(b) Python

Figure 4: Percent Correct (*i.e.*, matching) sketches (top-1 generated sketch) *vs.* ground-truth length

Table 2: Performance for GRAMMFORMER ablations (C#), for different $P_s$ and reward functions.

| | REGEXACC | | ROUGE | Avg Length |
|---|---|---|---|---|
| | Top 1 | Top 5 | | |
| GRAMMFORMER | 0.47 | 0.59 | 77.4 | 7.5 |
| Random expansion, no ⊘ | 0.42 | 0.54 | 78.3 | 8.1 |
| Random expansion, ⊘ at fixed threshold | 0.45 | 0.57 | 71.6 | 5.8 |
| GRAMMFORMER, $r(\cdot)$ = ROUGE$_{F1}$ | 0.42 | 0.54 | 78.2 | 8.1 |
| GRAMMFORMER, $r(\cdot)$ = REGEXACC | 0.51 | 0.62 | 70.8 | 5.8 |
| $L \rightarrow R$ | 0.42 | 0.47 | 77.0 | 7.1 |
| GRAMMFORMER, no ⊘ | 0.42 | 0.55 | 78.1 | 8.2 |

model is similar to our first ablation, but stops expansion when the probability of the generated $\mathbf{x}^{(t)}$ falls below a threshold. We choose this threshold on the validation set. This model makes shorter, but more accurate sketch predictions compared to the "random expansion", but is worse than GRAMMFORMER. These two ablations demonstrate that our learned $P_s$ is useful.

Second, we consider the effect of using different reward functions $r(\cdot)$ in the training of GRAMM-FORMER. Concretely, whereas we use the mean of REGEXACC and ROUGE in GRAMMFORMER, we now consider using only a single of these two metrics. As expected, the results on the corresponding metric improve, with a substantial cost in the other metric. Concretely, using only REGEXACC leads to significantly shorter predictions with a low ROUGE score. We believe that this is because REGEXACC is a strict metric, returning 0 if the sketch does *not* match, which leads to sparse rewards and makes the resulting model more conservative at expanding non-terminals.

Finally, to evaluate the benefit of the grammar-guided decoder, we consider a variant of GRAMM-FORMER that does not allow the introduction of ■ and instead has continue expansion until no non-terminals exist anymore. This variant can be compared to $L \rightarrow R$, which also cannot stop or introduce ■ tokens. Our ablation shows that there is substantial benefit in using grammar-guided decoding, leading both to longer predictions as well as more correct ones.

## 3.1 QUALITATIVE EVALUATION

Having observed the quantitative results, we now turn our attention to a qualitative look at the results and show some cherry-picked examples that illustrate desired and undesired behaviours of GRAMMFORMER and the baselines, where we also include the suggestions of the GitHub Copilot system GitHub (2021). Fig. 1 shows an example and eleven more are shown in Appx. A. Fig. 1 illustrates the importance of generating sketches instead of concrete sequences of terminal tokens: oftentimes, the code context does not provide sufficient information about the user's intent. Sketch-generating models can offer more informative suggestions given the partial intent.

Of course, GRAMMFORMER also makes mistakes. For example, GRAMMFORMER and $L \rightarrow R +$ ⊘ are sometimes "too" conservative (*e.g.* Fig. 15 in Appx. A) generating holes where $L \rightarrow R$ generates fully concrete completions. This suggests future research opportunities for better calibration of $P_s$.

Finally, a pure language modelling approach to code completion will always be insufficient. For example, user-defined types and rare APIs cannot be predicted by a language model, since'' the APIs cannot be known during training (Fig. 7 and Fig. 17 in Appx. A). Researching methods to scalably introduce information from static analyses and additional context may alleviate this.

## 4 Related Work

One of the successful applications of LMCs is code completion (Svyatkovskiy et al., 2019; Karampatsis et al., 2020). Transformer LMs have shown exceptional performance at the task being able to predict relatively long code sequences (Svyatkovskiy et al., 2020; Chen et al., 2021). Grammar-based code completion and generation has been researched with neural (Maddison & Tarlow, 2014; Yin & Neubig, 2017; Kim et al., 2021) and non-neural models (Bielik et al., 2016), always expanding the left-most, bottom-most non-terminal. In contrast to GRAMMFORMERs, all these models target the generation of *complete* code without the ability to create sketches. R3NN (Parisotto et al., 2017) generates only complete programs of a simple string transformation DSL but expands the non-terminal with the highest confidence, instead of the left-most, bottom-most one, similar to GRAMMFORMER. In contrast to the aforementioned models, GRAMMFORMER does *not* maintain an explicit tree representation but instead uses the sequences of leaves in the generation tree.

Sketch-like ideas appear in NLP such as the coarse-to-fine semantic parsing of Dong & Lapata (2018) and chat-bots of Shum et al. (2019). However, sketches are extracted deterministically to create a supervised dataset. Similarly, SketchAdapt (Nye et al., 2019) uses a sequence model to generate sketches for small functional programs of a simple DSL towards speeding-up enumerative program synthesis from input-output examples. SketchAdapt is also trained as a supervised sketch generator. A supervised corpus is created by enumerating all possible sketches and selecting the one with the highest-probability and within a heuristically computed time budget. In GRAMMFORMER domain, enumerating all sketches is computationally intractable due to the complexity of general-purpose programming languages while no similar heuristic exists for code completion.

Recently, sequence generation approaches beyond the left-to-right paradigm have been proposed (Welleck et al., 2019; Stern et al., 2019; Gu et al., 2019; Ford et al., 2018; Lee et al., 2018; Shah et al., 2018), usually by considering generation as an procedure that iteratively changes or extends a sequence. These models often aim in speeding-up inference or allowing models to figure a better order for full sentence generation. However, since these models focus on natural language and since its grammar is not defined a priori, these methods do not follow a grammar that limits the space for sketch generation. Additionally, these generate full utterances of text, rather than sketches.

A related concept is learning to abstain (Ziyin et al., 2019) where a model learns to predict a "don't know". This resembles the stop symbol "⊘" with the difference that GRAMMFORMER employs RL to learn $P_s$ for a sequential problem rather than learning to abstain for a single-step classification.

## 5 Discussion & Conclusions

In this work, we presented GRAMMFORMER, a generative model of code that goes beyond standard left-to-right generation and is able to generate sketches, *i.e.* snippets of code with holes. Designing generative machine learning models with such abilities is important towards facilitating better collaboration between machine learning models and their human users.

While we showed that GRAMMFORMER performs better than alternatives in sketch generation, there are still many research opportunities. First, larger transformers will most probably yield better results, as shown in the literature. Second, although we used REGEXACC as an evaluation metric, human studies to evaluate it are needed. Such studies, similar to those in machine translation and summarization, can yield more informed $r(\cdot)$ and improved user experiences. Second, although we focused on programming languages, modelling natural language also seems possible. Finally, we treated programming languages as a sequence of terminals and non-terminals, ignoring the structure imposed by code's semantics, *e.g.* data and control flow. Explicitly providing x'the code's structure, *e.g.* with relational transformers (Hellendoorn et al., 2019) may further improve GRAMMFORMER.

ACKNOWLEDGMENTS

The authors would like to thank Alex Polozov for useful discussions. We also thank Patrick Fernandes, Szymon Malik, and Guilherme Ilunga for working on earlier modeling ideas on sketch generation. Although those were unsuccessful, they provided the inspiration for this work.

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

## A   GENERATED SAMPLES

Fig. 5 and Fig. 6 show two examples from our dataset along with the ground-truth and the sequence of expansions performed by GRAMMFORMER. Fig. 7-13 show example generations by GRAMM-FORMER and the baseline models $L \rightarrow R$ and $L \rightarrow R + \oslash$. The parentheses in red indicate the REGEXACC score for each suggestion. For the $L \rightarrow R + \oslash$ baseline the special non-terminal `<suffix>` is added to indicate that a hole is introduced at the end of the left-to-right generation. Finally Fig. 15-17 show example generations where GRAMMFORMER make mistakes. A discussion for each of those sample is found at the caption of each figure.

```
Context:
using System.Collections;
using System.Collections.Generic;
using UnityEngine;
public class BottleFlipAcademy: Academy{
    public float MaxDistance;
    public float MinScale;
    public bool IsRandomDirection;
    public override void AcademyReset(){
        MaxDistance = resetParameters["max_distance"];
        <expression_statement>
```

```
Ground Truth:
MinScale = resetParameters["min_scale"];
```

```
Prediction:
MinScale = resetParameters[<string_literal>];
```

```
Generation Process:
<expression_statement>
<left> <assignment_operator> <right> ;
<left> = <right> ;
<identifier> = <right> ;
<identifier> = <expression> [ <argument> ] ;
<identifier> = <identifier> [ <argument> ] ;
<identifier> = resetParameters [ <argument> ] ;
<identifier> = resetParameters [ <string_literal> ] ;
MinScale = resetParameters [ <string_literal> ] ;
```

Figure 5: An example GRAMMFORMER generation for C#. Each line in the generation process shows subsequent states of $\mathbf{x}_t$ in Alg. 1. Here, GRAMMFORMER predicts a sketch that matches the ground-truth expansion, but places a hole at the key of the dictionary lookup, instead of predicting a low-likelihood string literal.

## B   DATASET STATISTICS

Some statistics about the datasets used throughout this work are shown in Tbl. 3

## C   FLATTENED NON-TERMINALS

The non-terminals in Tbl. 4 are always expanded and are not considered as non-terminals. Most of these non-terminals have always the same children (terminals or non-terminals), representing a single CFG rule. By flattening those non-terminals the depth of tree is reduced (and hence the number of loops needed in Alg. 1).

```
Context:
import sys
import os
import platform
if platform.system() == "Linux":
    os.system('clear')
elif platform.system() == "Windows":
    os.system('cls')
target = sys.argv[1]
<expression_statement>
```

```
Ground Truth:
ID = sys.argv[2]
```

```
Prediction:
<identifier> = sys.argv[2]
```

```
Generation Process:
<left> = <right>
<left> = <subscript>
<left> = <value> [ <subscript> ]
<identifier> = <value> [ <subscript> ]
<identifier> = <attribute> [ <subscript> ]
<identifier> = <object> . <attribute> [ <subscript> ]
<identifier> = <object> . <identifier> [ <subscript> ]
<identifier> = <object> . argv [ <subscript> ]
<identifier> = <object> . argv [ <integer> ]
<identifier> = <object> . argv [ 2 ]
<identifier> = <identifier> . argv [ 2 ]
<identifier> = sys . argv [ 2 ]
```

Figure 6: An example GRAMMFORMER generation for Python. Each line in the generation process shows subsequent states of $\mathbf{x}_t$ in Alg. 1. GRAMMFORMER here predicts that the user's intent is to read-in a second argument and store it in a variable. However, within the current context, the name of the variable storing the second argument would be impossible to predict. GRAMMFORMER— reasonably — places a hole at the given location and generates a matching sketch. In this example, any traditional left-to-right model would need to first predict an accurate target variable name (which seems unlikely in the given context) before predicting the right-hand side of the assignment.

|  | Python | C# |
| --- | --- | --- |
| Num Training Files/Trees | 1973400 | 1948516 |
| Num Validation Files/Trees | 218398 | 216299 |
| Num Test Files/Trees | 460874 | 480166 |
| Avg num tokens of $\mathbf{x}_t$ | 194.5 | 201.4 |
| Median num tokens of $\mathbf{x}_t$ | 205 | 206 |
| 99 percentile num tokens of $\mathbf{x}_t$ | 250 | 260 |
| Avg num tokens of $\mathbf{y}$ | 1.9 | 1.9 |
| Median num tokens of $\mathbf{y}$ | 1 | 1 |
| 99 percentile num tokens of $\mathbf{y}$ | 9 | 7 |

Table 3: Statistics of the datasets used.

## D   UNDERSTANDING REGEXACC

Since REGEXACC is a new metric, we include two deterministic ways of introducing sketches in Tbl. 5. First, if all literals (strings, numeric) are replaced with a hole, we see that a high REGEXACC

```
Context:
...
ExchangeActivity
        = (ExchangeActivity) Singleton<ActivitySys>.GetInstance().GetActivity(
    COM_WEAL_TYPE.COM_WEAL_EXCHANGE,
    msg.stPkgData.stWealExchangeRes.dwWealID) ;

if ( exchangeActivity != null ){
    exchangeActivity.IncreaseExchangeCount(
        (int) msg.stPkgData.stWealExchangeRes.bWealIdx,
        msg.stPkgData.stWealExchangeRes.dwDrawCnt );
    <expression_statement>
```

```
Ground Truth:
exchangeActivity.UpdateView();
```

**Prediction:**

$L \rightarrow R$:
```
Singleton<CUIManager>.GetInstance().CloseSendMsgAlert(); (0.000)
```

$L \rightarrow R + \bigcirc$:
```
Singleton<<suffix> (0.000)
```

$L \rightarrow R \cup \blacksquare$:
```
exchangeActivity.<hole>(); (0.833)
```

*CoPilot*:
```
} (0.000)
```

*GrammFormer*:
```
exchangeActivity.<identifier>(); (0.833)
```

Figure 7: A C# example and completion outputs from different models. REGEXACC score reported in red. Here, GRAMMFORMER correctly identifies that a method should be invoked on exchangeActivity, but does not predict the concrete method. If GRAMMFORMER was extended with information from a static analysis about the ExchangeActivity (potentially a user-defined type) then an accurate suggestion could have potential been made.

is achieved. In contrast, replacing both identifiers and literals (leaving "just" parentheses, brackets, dots, *etc.*) we get an easy "lower-bound". Note how C# — which is syntactically more verbose — achieves a better score, compared to Python. In Tbl. 6, we show some example sketches and their associated REGEXACC score.

## E    BEAM SEARCH

Alg. 2 presents the beam search used in GRAMMFORMER.

```
Context:
...
[Test] public void CanPassTwoProviders( ){
    // arrange
    var expectedLength = 100;
    var input1 = new TestSampleProvider(44100, 2, 50);
    var input2 = new TestSampleProvider(44100, 2, 50);
    var concatenator = new ConcatenatingSampleProvider(new[]{input1, input2});
    var buffer = new float[2000];
    var read = concatenator.Read(buffer, 0, buffer.Length);
    Assert.AreEqual(expectedLength, read, "read == expectedLength");
    Assert.AreEqual(49, buffer[49]);
    <expression_statement>
```

```
Ground Truth:
Assert.AreEqual(0, buffer[50]);
```

```
Prediction:
L → R:
Assert.AreEqual(50, buffer[50]); (0.000)

L → R + ⊘:
Assert.AreEqual(<suffix> (0.333)

L → R ∪ ■:
Assert.AreEqual(expectedLength, read); (0.00)

CoPilot:
Assert.AreEqual(50, buffer[50]); (0.000)

GrammFormer:
Assert.AreEqual(<argument>, buffer[50]); (0.917)
```

Figure 8: A C# example and completion outputs from different models. REGEXACC score reported in red. Here, GRAMMFORMER correctly predicts that an `AreEqual` assert statement should be made, checking the value of `buffer[50]`. However, within this context, the correct concrete expected value (`0`) would be hard to predict, even for a human. GRAMMFORMER places a hole there and generates a correct line-level sketch. In contrast, $L \rightarrow R$ introduces a wrong completion and $L \rightarrow R + \oslash$ creates a correct, but much shorter sketch.

---

**Algorithm 2** GRAMMFORMER beam search, given an input sequence $\mathbf{x}_0$.

$b \leftarrow \{(\mathbf{x}_0, 0, \texttt{false})\}$    $\triangleright$ Initialize Beam (state, logprob, isDone)
**while** $\exists (\mathbf{x}, p, isDone) \in b$ with $isDone = \texttt{false}$ **do**    $\triangleright$ While beam contains incomplete generations
     $b' \leftarrow \{\}$
     **for** $(\mathbf{x}, p, isDone) \in b$ **do**    $\triangleright$ For each sample in beam
         **if** isDone **then**    $\triangleright$ If suggestion is complete
             $b' \leftarrow b' \cup \{(x, p, isDone)\}$    $\triangleright$ No operation, beam is complete
             **continue**
         **for** $i \in \text{TOPM}(P_s(i|\mathbf{x}, N(\mathbf{x})))$ **do**    $\triangleright$ Get top-$m$ non-terminal positions
             $p_s \leftarrow \log P_s(i|\mathbf{x}, N(\mathbf{x}))$
             **if** $i = \oslash$ **then**
                 $b' \leftarrow b' \cup \{(x, p + p_s, \texttt{true})\}$    $\triangleright$ Stop Expansion
             **else**
                 **for** $\mathbf{y} \in \text{TOPN}(P_e(\mathbf{y}|\mathbf{x}, i))$ **do**    $\triangleright$ Beam search on $\mathbf{y}$ yields $n$ candidates
                     $p_e \leftarrow P_e(\mathbf{y}|\mathbf{x}, i)$
                     $b' \leftarrow b' \cup \{(\mathbf{x}_{<i} :: \mathbf{y} :: \mathbf{x}_{>i}), p + p_s + p_e, \texttt{false})\}$    $\triangleright$ Expand $x_i$
     $b \leftarrow \text{TOPK}(b')$    $\triangleright$ Prune Candidates and keep top $k$
**return** $b$

---

```
Context:
...
namespace Bug604053.Prueba{
    public class Data{
        public int M1{get; set;}
        public string M2{get; set;}
        public Data(int m1, string m2){M1 = m1; M2 = m2;}
        }
    [DataObject(true)] public class DataSource{
        public Data[] Retrieve( ){
            Data[] data = new Data[10];
            for (int i = 0; i<10; i++){
                <expression_statement>
```

```
Ground Truth:
data[i] = new Data(i, i.ToString());
```

```
Prediction:
L → R:
data[i] = new Data( ); (0.000)

L → R + ⊘:
data[i] = new Data( ); (0.000)

L → R ∪ ■:
data[i] = new Data( ); (0.000)

CoPilot:
data[i] = new Data(i, "Data" + i); (0.000)

GrammFormer:
data[i] = new Data(i, <string_literal>); (0.706)
```

Figure 9: A C# example and completion outputs from different models. REGEXACC score reported in red. While all models predict that an assignment needs to be made to each data[i], the exact form of the constructor is hard to predict. GRAMMFORMER seems to be looking at the constructor definition and predicts that some ⟨StringLiteral⟩ needs to be used as the second argument, although it is uncertain about its concrete form, hence introducing a hole.

```
Context:
# Provides a character-based width estimate when simple tags
# such as  and  are present in a multi-line,
# \"break\"-delimited, string. Very approximate, but a useful
# default.
def htmlWidth(sIn):
    iBr = indexOfBr(sIn)
    if (-1 == iBr):
        s = sIn
    else:
        s = sIn[:iBr]
    <return_statement>
```

```
Ground Truth:
return len(s)
```

**Prediction:**

$L \rightarrow R$:
```
return len(s) (1.000)
```

$L \rightarrow R + \oslash$:
```
return <suffix> (0.200)
```

$L \rightarrow R \cup \blacksquare$:
```
return s (0.000)
```

*CoPilot*:
```
s = s.replace(" "," ") (0.000)
```

*GrammFormer*:
```
return len(s) (1.000)
```

Figure 10: A Python example and completion outputs from different models. REGEXACC score reported in red. Here both $L \rightarrow R$ and GRAMMFORMER predict the full line correctly, but $L \rightarrow R + \oslash$ seems to return a more conservative (but correct) sketch.

```
Context:
#!/usr/bin/env python2
from __future__
import print_function
import argparse
import os
import subprocess
import sys
ap = argparse.ArgumentParser()
ap.add_argument("--release", action = "store_true")
ap.add_argument("--prerelease", action = "store_true")
<expression_statement>
```

```
Ground Truth:
ap.add_argument("--experimental", action = "store_true")
```

```
Prediction:
L → R:
args = ap.parse_args() (0.000)

L → R + ⊘:
args = ap.parse_args() (0.000)

L → R ∪ ■:
args = ap.add_argument(<hole> (0.000)

CoPilot:
ap.add_argument("--beta", action = "store_true") (0.000)

GrammFormer:
ap.add_argument(<string>, action = "store_true") (0.833)
```

Figure 11: A Python example and completion outputs from different models. REGEXACC score reported in red. See main text in the introduction for a description.

| | |
|---|---|
| Python | block, tuple, and, or, +, −, *, /, &, \|\|, //, %, @, +=, −=, *=, /=, //=, @=, &=, \|=, call, keyword_argument, name, binary_operator, for_in_clause, unary_operator, **, true, not_operator, none, false, boolean_operator, augumented_assignment, await, >>, pair, \|, parameters, <<, dictionary_comprehension, ellipsis, arguments, assignment, ^, ~ |
| C# | block, tuple, and, or, +, −, *, /, &, \|\|, //, %, @, +=, −=, *=, /=, //=, %=, @=, &=, \|=, **, >>, \|, <<, ^, ~, assignment_expression, invocation_expression, arguments, member_access_expression, try_statement, catch_clause, conditional_expression, ==, array_type, rank, base_expression, conditional_access_expression, member_binding_expression, initializer, null_literal, >, element_access_expression, subscript, ??, this_expression, implicit_array_creation_expression, cast_expression, !=, variable_declaration, implicit_type, &&, as_expression, as, <, local_declaration_statement, if_statement, >=, <=, throw_expression, default_expression, pattern, is_pattern_expression, binary_expression, bracketed_argument_list, name, object_creation_expression, await_expression, , |

Table 4: Non-terminals that are always expanded in the Tree-Sitter grammar for the two languages considered.

```
Context:
import sys
import os
import platform
if platform.system() == "Linux":
    os.system('clear')
elif platform.system() == "Windows":
    os.system('cls')
target = sys.argv[1]
<expression_statement>
```

```
Ground Truth:
ID = sys.argv[2]
```

```
Prediction:
```
$L \rightarrow R$:
```
target = target.replace("\\\\", "/") (0.000)
```

$L \rightarrow R + \oslash$:
```
target = <suffix> (0.000)
```

$L \rightarrow R \cup \blacksquare$:
```
print(target) (0.000)
```

*CoPilot*:
```
No Suggestion (0.000)
```

*GrammFormer*:
```
<identifier> = sys.argv[2] (0.875)
```

Figure 12: A Python example and completion outputs from different models. REGEXACC score reported in red. Generation steps of GRAMMFORMER shown in Fig. 6. $L \rightarrow R$ and $L \rightarrow R + \oslash$ cannot generate correct sketches since the first token would be impossible to guess within this code context.

|  | C# | Python |
|---|---|---|
| Replace all literals with holes | 0.865 | 0.608 |
| Replace all literals and identifiers with holes | 0.126 | 0.060 |

Table 5: REGEXACC when deterministically introducing holes at specific location.

Table 6: Example REGEXACC scores for a variety of sketches.

| Ground-truth | |
|---|---|
| `ap.add_argument("--experimental", action="store_true")` | |
|  | REGEXACC |
| `ap.add_argument(■, action="store_true")` | 0.9 |
| `ap.add_argument(■, action=■)` | 0.8 |
| `ap.add_argument(■, ■)` | 0.6 |
| `ap.add_argument(■, action="store_false")` | 0.0 |
| `ap.add_argument(■, required=■)` | 0.0 |

```
Context:
...
console.setLevel(logging.DEBUG)
log.addHandler(console)
# moving assembled contigs (scaffolds) to misc dir
if os.path.isfile(args.corrected):
    shutil.move(args.corrected, args.assembled)
tmp_dir_for_corrector = os.path.join (args.output_dir, "mismatch_corrector", args.assembly_type)
# correcting
result_corrected_filename = os.path.join(tmp_dir_for_corrector, "corrected_contigs.fasta")
<expression_statement>
```

```
Ground Truth:
dst_configs = os.path.join(tmp_dir_for_corrector, "configs")
```

```
Prediction:
L → R:
result_corrected_fasta = os.path.join(args.output_dir, "corrected_fasta") (0.000)

L → R + ⊘:
result_corrected_fasta = <suffix> (0.000)

L → R ∪ ■:
result_corrected_filename = <hole> (0.000)

CoPilot:
if os.path.isfile(tmp_dir_for_corrector): (0.000)

GrammFormer:
<identifier> = os.path.join(tmp_dir_for_corrector, <string>) (0.769)
```

Figure 13: A Python example and completion outputs from different models. REGEXACC score reported in red. GRAMMFORMER completes the line creating a correct sketch with two holes at locations avoiding to make the mistakes that $L \to R$ and $L \to R + \oslash$ makes.

```
Context:
...
left = right = new Rect(0, 0, config.borderSize, height);
right.x = width - config.borderSize;
top = bottom = new Rect(config.borderSize, 0, width-config.borderSize*2, config.borderSize);
bottom.y = height-config.borderSize;}
Color32 previousColor;
public override void OnGUI(){
    if(!props.wrongActionShowFrame)
        return;
    previousColor = GUI.color;
    GUI.color = config.borderColor;
    GUI.DrawTexture(left, config.texture);
    GUI.DrawTexture(right, config.texture);
    <expression_statement>
```

```
Ground Truth:
GUI.DrawTexture(top, config.texture);
```

```
Prediction:
L → R:
GUI.color = previousColor; (0.000)

L → R + ⊘:
GUI.color = previousColor; (0.000)

L → R ∪ ■:
GUI.color = Color.white; (0.000)

CoPilot:
GUI.DrawTexture(top, config.texture); (1.000)

GrammFormer:
GUI.DrawTexture(bottom, config.texture); (0.000)
```

Figure 14: A C# example and incorrect completion outputs from different models. REGEXACC score reported in red. The prediction from GRAMMFORMER is almost right but should have created a hole at the first argument for the user to fill-in. This shows that improved methods for training the policy network may improve results in the future.

```
Context:
...
namespace AspNetMvcCorePerformance{
    public class Program{
        public static int Main(string[] args){
            try {
                string urlBase = "http://localhost:54562/";
                var threadCount = 1;
                var iterationsPerThread = 50;
                if (args?.Length > 0){
                    urlBase = args[0];
                    threadCount = int.Parse(args[1]);
                    <expression_statement>
```

```
Ground Truth:
iterationsPerThread = int.Parse(args[2]);
```

```
Prediction:
L → R:
iterationsPerThread = int.Parse(args[2]); (1.000)

L → R + ⃠:
iterationsPerThread = int.Parse(args[2]); (1.000)

L → R ∪ ■:
iterationsPerThread = int.Parse(args[2]); (1.000)

CoPilot:
iterationsPerThread = int.Parse(args[2]); (1.000)

GrammFormer:
iterationsPerThread = <integer_literal>; (0.250)
```

Figure 15: A C# example and completion outputs from different models. REGEXACC score reported in red. GRAMMFORMER suggests a correct sketch but the right-hand side of the assignment has to stop expansion since ⟨IntegerLiteral⟩ cannot generate `int.Parse(args[2])`. This suggests some of the limitations that the grammar-based generation of GRAMMFORMER may have, especially for shorter sequences.

```
Context:
import multiprocessing
from os import getenv
bind = '127.0.0.1:8001'
<expression_statement>
```

```
Ground Truth:
workers = multiprocessing.cpu_count() * 3
```

```
Prediction:
L → R:
workers = 2 (0.000)

L → R + ⃠:
workers = <suffix> (0.222)

L → R ∪ ■:
workers = 4 (0.000)

CoPilot:
workers = multiprocessing.cpu_count() * 2 + 1 (0.000)

GrammFormer:
<identifier> = <string> (0.111)
```

Figure 16: A Python example and completion outputs from different models. REGEXACC score reported in red. Although the sketch of the prediction from GRAMMFORMER is typically correct, it is not useful. Researching better evaluation metrics may improve GRAMMFORMER.

---

**Context:**
```
# import python modules
import random
import time
import OSC
# Connect to SuperCollider's internal port
<expression_statement>
```

**Ground Truth:**
```
c = OSC.OSCClient()
```

**Prediction:**

$L \rightarrow R$:
```
OSC.ConnectCollider() (0.000)
```

$L \rightarrow R + \oslash$:
```
OSC.<suffix> (0.000)
```

$L \rightarrow R \cup \blacksquare$:
```
OSC.connect() (0.000)
```

*Copilot*:
```
client = OSC.OSCClient() (0.000)
```

*GrammFormer*:
```
conn = OSC.connect() (0.000)
```

---

Figure 17: A Python example and completion outputs from different models. REGEXACC score reported in red. All model fail to invoke the correct API of the library. A potential future direction to mitigate the problem is to incorporate definitions of the external or system classes.

