# OpenReview forum: "Learning to Complete Code with Sketches"
_ICLR.cc/2022/Conference — ICLR 2022 Poster_

### Official Review · Reviewer_Qkek · 2021-11-02

**Correctness:** 4
**Technical Novelty And Significance:** 3
**Empirical Novelty And Significance:** 3
**Recommendation:** 8
**Confidence:** 5

**Main Review:**

All three contributions of this paper are strong and interesting:
Grammformer - a model that can predict code but has the escape hatch of predicting a “hole token” whenever it is uncertain.
RegexAcc - a metric for evaluating the fit of predictions with holes. This is an important metric for the evaluation of code completion techniques in general. It is definitely not perfect, even for this task, but it is an interesting perspective going beyond simply matching tokens.
The evaluation shows that the approach outperforms existing techniques for code completion. It contains both quantitative and qualitative evaluation.

* You are using two separate models for non-terminal selection and non-terminal expansion but partially share their parameters, did you try it with completely separate models first? Can you report what was the impact of that? Is that even possible given that the pretraining scheme of P_s uses the weights from the shared encoder?

* In grammar flattening, can you comment on the avg/max depth of a tree before/after flattening? From table 4 it is hard to understand the magnitude of the flattening. Also, how important is this part in practice?

* In Table 2, the performance of L->R and Grammformer with no “stop expansion” is actually quite good, and generates avg length of 8.2 for the latter. I guess that because it does not use the P_s model, it can also run more efficiently? Isn’t that by itself a good practical compromise?

* Can you share any data about the location of “holes” in generated examples? How often is the “hole” in the first parts of the sentence, and how often is it close to the end? I think it would be interesting to see the ability of Grammformer to recover from “early holes”.

* Is Grammformer with no “stop expansion” conceptually equivalent to AnyCodeGen by Alon et. al. (ICML 2020)?

* Do you have any hypothesis on cases where the model is able to recover despite having holes early in the generation process? Does it require some “synch” symbol that breaks the sentence and allows the model to resynch with a sub-sentence distribution, as common in error recovery of parsers for example? Your qualitative evaluation seems to hint towards that, where generation after holes benefits from “synch” symbols such as “=” that break a statement.

* If recovery after holes depends on some “synch” symbols being present, it is reasonable to assume that L->R with masking of holes (that you suggest as a baseline) would perform better. It would help if you can show some cases where L->R with holes is not able to recover but Grammformer recovers properly.

*  Are matches like the one in figure 16 counted in “top k” match score? I guess they are, as there is 1 terminal present. How much of the overall Grammformer score is obtained with trivial expansions as in Figure 16? What happens if you exclude this kind of trivial expansion from being counted?

* Inference with this model seems rather expensive. Can you share some data on the inference times of the different models used in the paper?

Minor:
Table 4 was rather hard to find in the appendix

Some missing related work:
* Rabinovich, M., Stern, M., and Klein, D. Abstract syntax networks for code generation and semantic parsing, ACL 2017
* Brockschmidt et al., Generative Code Modeling with Graphs, ICLR 2019
* Amodio et al., Neural attribute machines for program generation, 2017
* Murali et al., Neural sketch learning for conditional program generation, ICLR 2018
* Chen et al., Tree-to-tree neural networks for program translation, NeurIPS 2018


**Summary Of The Paper:**

The paper presents a new model for code completion which allows the model to completions with “holes” that are inserted in places where the model is uncertain. The idea of generating “holes” to enable skipping over “hard parts” of the prediction is novel and interesting. To realize this idea, the authors present a model that generates partial syntax trees where some of the non-terminals may be left without further expansion.
The model is evaluated on C# and Python programs and is shown to outperform existing techniques. The also paper presents a thorough ablation study.


**Summary Of The Review:**

The paper presents an original approach to code completion which allows to predict “holes” where the model is not certain about the prediction and keep prediction going after the “hole”. Training a model that is able to generate completions past “hole tokens” is challenging, and the authors present an elegant and clever technique for doing that, based on reinforcement learning.
Overall, a nice original idea and nice execution. I think this paper should be accepted.

---

> ### Author Response · Authors · 2021-11-10
> **Review Response (Part 1/2)**
>
> Thank you for your kind review and detailed questions. [Reply split into two posts to avoid hitting the char limit]
>
> > You are using two separate models for non-terminal selection and non-terminal expansion but partially share their parameters, did you try it with completely separate models first? Can you report what was the impact of that? Is that even possible given that the pretraining scheme of P_s uses the weights from the shared encoder?
>
> We did _not_ try using two separate models so far.
>
> As you note, the pretraining scheme does not make this straightforward, though we could imagine a scenario in which the encoder model is shared for pre-training, and then during fine-tuning gets copied, so that the $P_s$ and $P_e$ models could develop independently.
> However, we do not believe this to be very promising and problematic given concerns about overall computational cost: having two separate models for $P_s$ and $P_e$ would roughly double the time needed to encode the input, which would be quite costly.
>
> > In grammar flattening, can you comment on the avg/max depth of a tree before/after flattening? From table 4 it is hard to understand the magnitude of the flattening. Also, how important is this part in practice?
>
> * The avg depth of a tree before/after flattening for C# is 8.3/3.1 and for Python is 8.1/4.8.
> * The max depth of a tree before/after flattening for C# is 127/93 and for Python is 163/65.
>
> We don't believe that flattening changes the quality of the predictions of Grammformer. However, it reduces the computational time during inference, since fewer expansions are necessary.
>
> > In Table 2, the performance of L->R and Grammformer with no “stop expansion” is actually quite good, and generates avg length of 8.2 for the latter. I guess that because it does not use the P_s model, it can also run more efficiently? Isn’t that by itself a good practical compromise?
>
> It seems that there are two questions contained here:
> 1. The relative performance of $L \to R$ and $\text{Grammformer, no}\ \obslash$.
>    The latter model is forced to always expand all non-terminals (and indeed, doesn't require a $P_s$ submodel). As you observe, it performs quite well - one part that we wanted to illustrate here was that grammar-guidance is providing a substantial boost over the sequence-based $L \to R$ model.
>    However, the lack of a facility to insert holes means that the model is forced to "hallucinate" contents even when very uncertain, so that for example in Fig. 1 of our paper, it would need to choose a variable name without having any knowledge about the semantics.
> 2. The runtime performance of the $\text{Grammformer, no}\ \obslash$ model.
>    Here, we would note that the $P_s$ model itself is computationally not particularly complicated - it's really just a small feedforward network operating on top of the representations of non-terminal tokens. The computationally expensive part is the repeated encoding of the partially generated output with the Transformer encoder, which this model still requires.
>
>
> > Can you share any data about the location of “holes” in generated examples? How often is the “hole” in the first parts of the sentence, and how often is it close to the end? I think it would be interesting to see the ability of Grammformer to recover from “early holes”.
>
> Focusing on the predictions with a single hole inserted, we see that on average there are:
> * C#: 5.3 token _before_ the hole and 3.1 tokens _after_ the hole.
> * Python: 1.6 tokens _before_ the hole and 2.2 tokens _after_ the hole.
>
> In practice, by empirically looking at the suggestions, it seems that Grammformer learns to place tokens either towards the beginning or towards the end of a suggestion.

---

> ### Author Response · Authors · 2021-11-10
> **Review Response (Part 2/2)**
>
> [Part 2 of our reply]
>
>
> > Is Grammformer with no “stop expansion” conceptually equivalent to AnyCodeGen by Alon et. al. (ICML 2020)?
>
> Not quite. There are two substantial differences:
>  1. AnyCodeGen uses (code) paths (as in code2vec and code2seq) to embed (partial) programs, whereas Grammformer uses a Transformer model operating directly on the token sequence (with no knowledge of the AST in the encoder).
>  2. AnyCodeGen generates one AST node (or subtoken) at a time, requiring re-encoding of the partial AST after each step, whereas Grammformer generates all children of a non-terminal AST node in one go before encoding the partial output again.
>
> However, there are number of commonalities. In particular, the idea of using non-terminal nodes (and their repeated expansion) of AST nodes to guide generation, without being restricted to specific grammar rules, is shared between Grammformer and AnyCodeGen.
>
> > Do you have any hypothesis on cases where the model is able to recover despite having holes early in the generation process? Does it require some “synch” symbol that breaks the sentence and allows the model to resynch with a sub-sentence distribution, as common in error recovery of parsers for example? Your qualitative evaluation seems to hint towards that, where generation after holes benefits from “synch” symbols such as “=” that break a statement.
>
> It's important to note that the grammar-driven nature of Grammformer means that there is no notion of holes that exist "early" in the generation procedure. As an example, consider Fig. 2 in our submission: as you can see, non-terminals (or "holes") occur in all steps of the generation procedure. This also becomes visible in the generation sequences illustrated in Figs. 5 and 6. We found the intuition of viewing Grammformer as a refinement procedure helpful, in which each step refines the non-terminal about which the model is least uncertain.
> The relation to "synch" symbols are then just an artefact of the fact that it's often easy to predict the use of production rules introducing such symbols (such as those producing assignment statements (for `=`) or method calls (for `.`)).
>
> > Are matches like the one in figure 16 counted in “top k” match score? I guess they are, as there is 1 terminal present. How much of the overall Grammformer score is obtained with trivial expansions as in Figure 16? What happens if you exclude this kind of trivial expansion from being counted?
>
> All partial matches are accounted in the average RegexAcc of Table 1. For example, the snippet in Fig. 16, has a RegexAcc of 0.111 and this value is averaged among the other suggestions to give the "Top-K" RegexAcc in Table 1.
> In this case, this example is below the average RegexAcc reported for Python.
>
> Note that although the completion is trivial, the model has "understood" that an assignment is necessary at this location, which isn't necessarily non-trivial. So, it's hard to explicitly classify (parts of) suggestions as trivial and non-trivial. Certainly, a portion of RegexAcc is due to the prediction of syntatic/punctuation parts: note how the differences between C# and Python show this. C# has a more verbose syntax (braces, parentheses, _etc._) and all models achieve better RegexAcc.
>
> We would love to hear your thoughts on better ways to measure the quality of the suggested sketches.
>
>
> > Inference with this model seems rather expensive. Can you share some data on the inference times of the different models used in the paper?
>
> Please see our general response for more details on this.
>
> > Some missing related work
>
> Thank you for bringing up these works. We will update the text to discuss them in the related work section and cite them appropriately.

---

### Official Review · Reviewer_suKF · 2021-11-03

**Correctness:** 4
**Technical Novelty And Significance:** 3
**Empirical Novelty And Significance:** 3
**Recommendation:** 6
**Confidence:** 4

**Main Review:**

**Strengths**

* The problem is very relevant from usability perspective of code completion. If the model is uncertain about certain code fragments, it makes sense to seek input from the user. The paper has nicely motivated the problem with a good example in Fig. 1.

* The idea of generating program sketches has been explored in literature in different contexts. ([2, 3] already mentioned in the paper. Also see[1]). However, to my knowledge, this is the first work where sketch generation is used in the context of code completion to prevent generation of code with low certainty.

* The formulation is interesting and quite different from the way language models are usually used to generate code. Instead of directly generating the code, language models are used to expand a non-terminal in the code. A separate model built to identify the non-terminal to expand. RL is employed as it is difficult to generate supervised data for this setting.

* The paper introduces REGEXACC measure. This is quite strict measure as its value of zero for all the sketches that can not be expanded to the ground truth. This measure can however be used in conjunction with other measures (as done in this paper).

**Weaknesses/Suggestions/Questions**

* Has other measures apart from ROUGE been considered for evaluating the sketch? There are several other measures that have been used to approximate the notion of program equivalence. (e.g. [1] uses Jaccard distance and various AST based measures). It would be nice to discuss these measures and contrast them with ROUGE and REGEXACC.

* Section 2.2 says, "simple extended sequence model from above do not perform well ...hole tokens would not replace semantically meaningful subsequences" This is a bit surprising considering the success of language models in generating grammatically correct sentences. The pretraining as described in the paper replaces AST non-terminals of target output by holes. The poor performance can be due to lack of sufficient data though.

* pg 4. "P_e is not directly constrained to follow the production rules." I am wondering why you didn't choose to train a probability distribution over the grammar rules. This will improve the results since the model will always generate syntactically correct programs. Is it for handling non-terminals like strings literals? (Could you possibly use a language model only for expanding non-terminals like String Literals.)

* Fig 1 has a row for CoPilot, but the it is not discussed in the main paper.

* "Evaluation > Baselines" says "we consider two transformer baselines ...". However, the Table 1 as well as the Fig. 4  have three transformer based baselines. "L -> R with holes" is not described in the paper. The "result" paragraph also talks only about "L -> R with stop policy". It would be nice to summarize all the baselines at one place.

* GRAMMAFORMER improves only marginally over GRAMMAFORMER (pre-trained only). This makes me wonder if it is worth brining in RL. I would like the authors to comment on this.

* [1] Murali, Vijayaraghavan, et al. "Neural Sketch Learning for Conditional Program Generation." International Conference on Learning Representations. 2018.
* [2] Li Dong and Mirella Lapata. Coarse-to-fine decoding for neural semantic parsing. In Proceedings of the 56th Annual Meeting of the Association for Computational Linguistics (ACL), 2018.
* [3] Nye, M., Hewitt, L., Tenenbaum, J., and Solar-Lezama, A. Learning to infer program sketches. In International Conference on Machine Learning, pp. 4861–4870, 2019.

**Summary Of The Paper:**

This work proposes, GRAMMAFORMER, a transformer model for generating code with "holes" inserted in places where a model is uncertain. GRAMMAFORMER is trained on code completion task for C# and Python. The model generates 10-50% more accurate completions and 37-50% longer sketches.

**Summary Of The Review:**

This work brings in the several techniques (Grammar based program synthesis, Language models over partial programs, separate model for non-terminal selection, etc ) to solve the problem of code completion with sketches. I would rate the paper medium on novelty as many of these techniques have been already explored in the literature in different contexts. However, to me, the important contribution of this paper is the problem setting. In my opinion, Keeping placeholders in place of low-probability program fragments can improve the utility of code completion tools to a great extent.

---

> ### Author Response · Authors · 2021-11-10
> **Review Response**
>
> Thank you for your detailed review and the references to related work.
>
>
> > Has other measures apart from ROUGE been considered for evaluating the sketch? There are several other measures that have been used to approximate the notion of program equivalence. (e.g. [1] uses Jaccard distance and various AST based measures).
>
> We consider RegexAcc and ROUGE in this work and use reinforcement learning to optimize directly on those metrics. Although these metrics might not be optimal, we believe that they reflect the code completion use case, where small mistakes are distracting and potentially harmful. Using additional metrics during training and testing is possible, but we have not experimented with them. Finally, as we discuss in the paper, we believe that performing user surveys to better reflect on the quality of each suggestion is important future work.
>
> > Section 2.2 says, "simple extended sequence model from above do not perform well ...hole tokens would not replace semantically meaningful subsequences" This is a bit surprising considering the success of language models in generating grammatically correct sentences.
>
> Note that the models succeed in generating meaningful sequences, but the produced holes are not necessarily meaningful to a human. One example is given in the paper, where `szconv.<HOLE>)` is in principle a reasonable suggestion, but requires replacing `<HOLE>` by `method(` (note that this needs to include the `(`). We will update the text to reflect this.
>
> > pg 4. "P_e is not directly constrained to follow the production rules." I am wondering why you didn't choose to train a probability distribution over the grammar rules. This will improve the results since the model will always generate syntactically correct programs. Is it for handling non-terminals like strings literals?
>
> Indeed, handling literals is one case that is improved by this choice; more common however are tuples of different arity (e.g., the argument tuple for method calls).
>
> In general, all cases in which the language grammar allows an arbitrary number of non-terminals become easier. For example, for Python, you can see the large number of related symbols on https://docs.python.org/3/library/ast.html, looking for occurrences of things such as `expr*`, `arg*`, etc.
>
> > "Evaluation > Baselines" says "we consider two transformer baselines ...". However, the Table 1 as well as the Fig. 4 have three transformer based baselines. "L -> R with holes" is not described in the paper.
>
> This was an editing mistake shortly before submission time. As discussed in our general response, we have now included the additional baseline results, which were not complete in time for the ICLR deadline.

---

### Official Review · Reviewer_Arrw · 2021-11-04

**Correctness:** 3
**Technical Novelty And Significance:** 3
**Empirical Novelty And Significance:** 3
**Recommendation:** 5
**Confidence:** 4

**Main Review:**

Strengths
------------
- Explores an interesting and potentially useful variant of code completion
- The evaluation shows that the proposed method outperforms simple baselines like the unmodified transformer.

Weaknesses
------------------
- The gain in performance over the standard transformer is small according to Table 1
- it is not clear how much computational overhead Grammformer adds over the standard transformer during inference. I could not find any experimental results showing the overhead.
- I also do not quite follow why metrics like ROUGE and REGEXACC are good fits for code completion tasks. The authors provide no evidence that these metrics derived from natural language tasks are meaningful for programming languages too.

Overall, I find the idea of sketch generation to be potentially useful. The authors described the proposed algorithm clearly. However, my main concern with this paper is that the empirical evidence demonstrating the quality/usefulness of the generated sketches (for programming language-related tasks) is rather thin. For example, what are the downstream tasks the authors envision the generated sketches to be useful for? If it is to help the developers by providing coding templates, it is not clear to me that ROGUE or REGEXACC are the right metrics to measure the usefulness of the sketches. I think an end-to-end evaluation of the usefulness of the sketches for a specific application would have been much more convincing.



**Summary Of The Paper:**

The problem of code completion is often hard because some intermediate strings might be difficult to predict while the nearby tokens might still be easy to predict. To exploit this observation, this paper presents Grammformer, a transformer-based model for generating code completions with "holes" inserted in places where the model is uncertain.

**Summary Of The Review:**

Interesting idea and well-written paper. However, without end-to-end evaluation on specific tasks, it is hard for me to judge the usefulness of the generated code sketches.

---

> ### Author Response · Authors · 2021-11-10
> **Review Response**
>
> Thank you for engaging with our submission.
>
> > The gain in performance over the standard transformer is small according to Table 1
>
> We respectfully disagree. In practice, we believe the "Top-1 RegexAcc" metric to be most representative of user experience (in that this is the likelihood that the first autocomplete proposal can be completed to the correct result, and is as precise as possible). On that metric, our model is a 24% relative improvement on Python and a 12% improvement on C#.
> If users are willing to look at further proposals (e.g., the Top-5 metric), these improvements grow to 50% and 26%, which are clearly highly relevant.
>
> > it is not clear how much computational overhead Grammformer adds over the standard transformer during inference. I could not find any experimental results showing the overhead.
>
> Please see our general response for more details on this.
>
> > I also do not quite follow why metrics like ROUGE and REGEXACC are good fits for code completion tasks. The authors provide no evidence that these metrics derived from natural language tasks are meaningful for programming languages too.
>
> As we note in the paper, we believe ROUGE not to be a good metric for these completion task - hence we developed the RegexAcc metric, which is specialized to the programming language context in that it puts more emphasis on correctness (in that examples with small mistakes yield 0 on this metric).
>
> > For example, what are the downstream tasks the authors envision the generated sketches to be useful for?
>
> As discussed in the paper (see abstract and introduction), we aim at the end-user scenario of providing code completion suggestions in the editor.

---

### Official Review · Reviewer_aRD4 · 2021-11-09

**Correctness:** 4
**Technical Novelty And Significance:** 3
**Empirical Novelty And Significance:** 3
**Recommendation:** 8
**Confidence:** 4

**Main Review:**

I really enjoyed reading this paper, and I think the authors devised a very promising approach to overcome the fundamental limitations of left-to-right LMC. Additionally, the paper is very well-written, and the discussion of related work is exhaustive.

STRENGTHS:
- the motivation in Fig. 1 is crystal clear, as it highlights a fundamental limitation of left-to-right LMC on generative tasks. The provided context is not enough to make a reliable prediction of the identifier name, so the model seeks the user input. This is both an effective and elegant solution which I hope will be adopted by future AI-driven autocompleters.
- to my knowledge, Grammformer is the first grammar-guided model that can generate code around "holes" that are added to the sequence in order to skip tokens with high uncertainty
- the combination of RL and a novel metric (RegexAcc) to overcome the lack of supervised data

WEAKNESSES:
- my main concern is about the runtime performance. As the author states in 2.2 "Computational Cost", there is an additional computational cost compared to a standard transfomer encoder-decoder model. From my understanding (which should be stated explicitly in the text), the inference cost grows linearly with the length of the predicted sequence, thus making this approach possibly unsuitable for autocompletion in the IDE (as the latency would be unacceptable for the user).
Rather than leaving this exercise to the reader, the paper should report clearly what are the runtime performance tradeoffs of Grammformer vs other approaches (especially considering the fact that GitHub copilot is among the compared systems).

- to exacerbate the aforementioned issue, it is worth noticing that the accuracy improvement brought by Grammformer is not astounding (hence the performance tradeoff could be considered even more disadvantageous). On a similar note, the pre-trained only version of Grammformer performs almost as good as the full-fledged model, but with potentially a much lower computational cost.

MINOR ISSUES:
- CoPilot appears in several figures, but it is not referred at all in the text. Please fix accordingly.
- I can't understand the sentence "x is the 200 terminal tokens" on page 6. Please clarify.

**Summary Of The Paper:**

This paper presents a novel approach to complete code via sketches, i.e., the proposed model can opt to leave a "hole" in the prediction whenever it is highly uncertain about the output. The experiments provide convincing arguments on why it is worth to adopt Grammformer when the prediction accuracy in the generated code is the main priority.

**Summary Of The Review:**

Grammformer is a thoughtful combination of novel contributions and familiar insights in the ML4code field.
While the runtime performance is unclear, the proposed approach can still deeply impact the future of LMC (especially for generative tasks), therefore I recommend this paper for appearing at ICLR in order to get the exposure it deserves.

---

> ### Author Response · Authors · 2021-11-10
> **Review Response**
>
> Thank you for your kind review.
>
>
> > my main concern is about the runtime performance
>
> Please see our general response for more details on this.
>
> > CoPilot appears in several figures, but it is not referred at all in the text. Please fix accordingly.
>
> Indeed, we have included GitHub Co-Pilot in the example figures to illustrate qualitatively what a very large, well-trained language model can achieve, but have not discussed this in the main text. We will update the text to clarify this.
>
> > "x is the 200 terminal tokens" on page 6
>
> In the paper, $\mathbf{x}$ canonically refers to the "input"/"prompt" of our method, see beginning of Sect. 2. Hence, what this means here is that all models get 200 tokens as input (before BPE is applied), and then have to propose a code completion.

---

### Author Response · Authors · 2021-11-10
**General Response**

We thank all the reviewers for their time, feedback, and thoughtful comments. We reply here to questions raised in several reviews, and include information about updates in our newest revision. Responses to individual comments are posted as replies to each review.

* **References to Copilot in figures**

  We have included GitHub Copilot results in the example figures to illustrate qualitatively what a very large, well-trained language model can achieve, but have not discussed this in the main text. Due to limited access, we cannot provide quantitative results for Copilot. We have updated the text of the paper to clarify this.

* **Additional Language Model baseline**

  We have updated the paper to additionally incude a standard (causal) language model beyond the encoder-decoder models we are already presenting.

  This model uses a single Transformer with 12 layers (so that it has the same number of parameters as the other models) and is trained in a similar fashion as the rest of the baselines.
  In particular, we consider the language model case, an extension with the ability to stop generation, and a second extension that is trained to insert holes.
  Overall, (pre)training is performing similar to our existing $L \to R$ baseline.

  The results indicate that Grammformer outperforms this baseline. Again, we believe that this is due to the hardness introduced by coupling the decision of deciding where to introduce a hole and where to continue generating concrete code tokens.

* **Computational Performance**
  | Model                    |         C#         |       Python       |
  | ------------------------ | :----------------: | :----------------: |
  | $L \to R$                | 0.3809 sec/example | 0.1999 sec/example |
  | $L \to R + \obslash$     | 0.2477 sec/example | 0.1108 sec/example |
  | $L \to R + \blacksquare$ | 0.2576 sec/example | 0.1273 sec/example |
  | Grammformer              | 0.6992 sec/example | 1.2869 sec/example |

  As can be seen, the runtime performance of Grammformer isn't prohibitive (roughly doubling the $L \to R$ baseline in C#, and leading to a 6x slow-down for Python due to deeper syntax trees and more non-terminals that require more expansion steps). We agree with the reviewers that future work should focus on reducing the computational cost. This could be achieved by using some of the efficient Transformer variants, or reducing the number of expansion steps, for example by expanding several non-terminals in parallel.

  We will include this information in a future version of the paper.

---

### Decision · Program_Chairs · 2022-01-20

**Decision:**

Accept (Poster)

**Comment:**

The paper proposes a transformer model of code that leaves "holes" at points of generation at which the model is uncertain. The model is evaluated on C# and Python programs and outperforms existing techniques.

The reviewers found the Grammformer model and the RegexAcc evaluation metric to be useful and interesting. The experimental results are also compelling. Given this, I recommend acceptance. Please make sure to incorporate the feedback in the reviews and the additional experimental results into the final version.